# Blue emission at atomically sharp 1D heterojunctions between graphene and h-BN

Gwangwoo Kim [1,11], Kyung Yeol Ma[2,11], Minsu Park[3], Minsu Kim[1], Jonghyuk Jeon [4], Jinouk Song[5], José Eduardo Barrios-Vargas [6], Yuta Sato [7], Yung-Chang Lin[7], Kazu Suenaga [7], Stephan Roche[8,9], Seunghyup Yoo[5], Byeong-Hyeok Sohn[4], Seokwoo Jeon[3] & Hyeon Suk Shin [1,2,10 ✉]

Atomically sharp heterojunctions in lateral two-dimensional heterostructures can provide the narrowest one-dimensional functionalities driven by unusual interfacial electronic states. For instance, the highly controlled growth of patchworks of graphene and hexagonal boron nitride (h-BN) would be a potential platform to explore unknown electronic, thermal, spin or optoelectronic property. However, to date, the possible emergence of physical properties and functionalities monitored by the interfaces between metallic graphene and insulating h-BN remains largely unexplored. Here, we demonstrate a blue emitting atomic-resolved heterojunction between graphene and h-BN. Such emission is tentatively attributed to localized energy states formed at the disordered boundaries of h-BN and graphene. The weak blue emission at the heterojunctions in simple in-plane heterostructures of h-BN and graphene can be enhanced by increasing the density of the interface in graphene quantum dots array embedded in the h-BN monolayer. This work suggests that the narrowest, atomically resolved heterojunctions of in-plane two-dimensional heterostructures provides a future playground for optoelectronics.

[1] Department of Chemistry, Ulsan National Institute of Science and Technology (UNIST), Ulsan 44919, Republic of Korea. [2] Department of Energy Engineering, Ulsan National Institute of Science and Technology (UNIST), Ulsan 44919, Republic of Korea. [3] Department of Materials Science and Engineering, Korea Advanced Institute of Science and Technology (KAIST), Daejeon 34141, Republic of Korea. [4] Department of Chemistry, Seoul National University, Seoul 08826, Republic of Korea. [5] School of Electrical Engineering, Korea Advanced Institute of Science and Technology (KAIST), Daejeon 34141, Republic of Korea. [6] Departamento de Física y Química Teórica, Facultad de Química, UNAM, 04510 México City, México. [7] Nanomaterials Research Institute, National Institute of Advanced Industrial Science and Technology (AIST), 1-1-1 Higashi, Tsukuba 305-8565, Japan. [8] Catalan Institute of Nanoscience and Nanotechnology (ICN2), CSIC and The Barcelona Institute of Science and Technology, Campus UAB, 08193 Barcelona, Spain. [9] ICREA—Institució Catalana de Recerca i Estudis Avançats, 08010 Barcelona, Spain. [10] Low Dimensional Carbon Material Center, Ulsan National Institute of Science and Technology (UNIST), Ulsan 44919, Republic of Korea. [11] These authors contributed equally: Gwangwoo Kim, Kyung Yeol Ma. ✉email: shin@unist.ac.kr

n-plane two-dimensional (2D) heterostructures serve as a perfect platform to explore one-dimensional (1D) functionalities because of their atomically sharp (resolved) heterojunctions. Because the edge contacts offer easier band offset tuning, lateral heterostructures-based heterojunctions have led to improved performance for P-N rectifying diode[1,2] and interlayer excitons[3], as well as low contact resistance for metal/semiconductor junctions[4,5]. These examples illustrate the emergence of quantum phenomena arising from differences in the properties of two types of 2D materials divided by a heterojunction. However, it has been rarely reported that 1D heterojunctions themselves show unknown properties and functionality.

Interestingly, a recent theoretical paper reported the formation of localized density of states (LDOS) at the random, disordered interface of hexagonal boron nitride (h-BN) and graphene[6]. Thus far, most of the theoretical calculations predicted that the zigzag and armchair interfaces between graphene and h-BN have half-metallic and semiconducting properties[7–9]. In addition, some experimental attempts using scanning tunneling microscopy have identified a clear sharp peak close to zero bias, which was ascribed to the localized interfacial states at the zigzag edges connecting graphene with h-BN[10,11]. Park et al. observed an enhanced local density of states at zigzag interfaces composed of several segments forming 120° angles and found distinct states with energy levels of −0.6 eV (or +0.6 eV) at C–B (or −N) terminated boundaries[12]. Finally, disordered interfaces between h-BN and graphene, which would be common in lateral heterostructures fabricated by patterned regrowth[13,14], have been observed experimentally, but the presence and role of localized energy states occurring at these interfaces are unknown.

Here, we demonstrate luminescent, atomic-resolved heterojunctions connecting graphene and h-BN domains in 2D in-plane heterostructures. Blue-emitting photoluminescence (PL) at 410 nm wavelength from 1D heterojunctions of h-BN and graphene was observed, which supposedly originates from the localized energy states at the disordered heterojunction between h-BN and graphene (G/h-BN junction). The weak blue emission at the 1D heterojunctions in a simple in-plane heterostructure of h-BN and graphene can be enhanced by increasing the amount of interface through embedding a graphene quantum dot array in a h-BN monolayer (GQD/h-BN in-plane heterostructures). Finally, we stacked four GQD/h-BN films by placing h-BN intercalation layer between each film to improve the emission.

## Results

### Simple in-plane graphene/h-BN heterostructures

First, we examine simple in-plane graphene/h-BN heterostructures to confirm that the desired heterostructures has formed[15]. Figure 1a shows an SEM image of the circular graphene region embedded in the h-BN monolayer, confirming a clear boundary between the graphene and h-BN regions. The boundary can be observed in a magnified SEM image (Fig. 1b). To identify the graphene and h-BN materials, we measured the Raman spectra of the heterostructures after they were transferred onto a SiO$_2$/Si substrate. The red and blue lines in Fig. 1c show the corresponding Raman spectra of the graphene (G, 2D peaks) and h-BN (E$_{2g}$ peak), respectively. Notably, the Raman mapping image (Fig. 1e) in the 2D band (2630–2730 cm$^{-1}$) clearly differentiates the graphene from the h-BN regions of the heterostructures. Figure 1d shows the PL spectra of the graphene (red), h-BN (blue), and the interface between the graphene and h-BN regions (green) by excitation with a 266 nm laser. Interestingly, we observed a small PL peak at 410 nm wavelength corresponding to the interface, while no PL emission was observed in the graphene or h-BN regions. Furthermore, PL

image mapping (Fig. 1f) at the 410 nm wavelength demonstrates the homogenous PL emission only at the circular shaped interface between the graphene and h-BN regions. The magnified mapping images (Supplementary Fig. 1) clearly show the luminescence occurring exclusively at the boundary. This PL result is different from the previous results for the graphene-h-BN zigzag interface that showed localized energy states near the Fermi level[10–12]. Since the grown in-plane heterostructures were fabricated in sequential growth, they resulted in epitaxial hetero-junctions between graphene and h-BN with well-defined zigzag edges. However, in our work, we fabricate heterostructures by using conversion reactions of h-BN to graphene and hence expect that a different atomic arrangement of the heterojunctions emerges at boundaries.

### Graphene quantum dots embedded in a h-BN monolayer

In order to increase the amount of interface per unit area for improved PL intensity, we prepare in-plane heterostructures of graphene quantum dots in a h-BN monolayer by a spatially controlled conversion on Pt nanoparticles (Pt NPs) prepared with the aid of self-patterning diblock copolymer micelles[16] (GQD/h-BN, Fig. 2a, b). The structure of GQD/h-BN is a better sample to investigate in-depth the PL emission at the G/h-BN junctions owing to the higher density of interfaces. Figure 2c presents the PL spectrum (red line) of the 7 nm-sized GQD/h-BN heterostructures under excitation with a 266 nm laser. The blue PL emission was observed with a peak wavelength of 410 nm, equivalent to the PL result (blue line) of the G/h-BN interfaces. Interestingly, a ~6 times enhancement in PL intensity is shown in GQD/h-BN films due to the higher interfacial density compared to the G/h-BN samples. The interface length per unit area for GQD/h-BN (0.0082 nm$^{-1}$) is ~6.3 times longer than that for the G/h-BN sample (0.0013 nm$^{-1}$), which is comparable to the ratio of PL intensities of the two samples.

To understand the mechanism of blue PL emission at 410 nm, we performed PL measurements on several control samples. It is noted that no PL peaks (Fig. 2c) were observed from bare Pt NPs (green) and bare h-BN (black) on a SiO$_2$/Si substrate. However, carbon doping in the h-BN monolayer during the conversion reaction might be the origin. To check, we achieved the conversion of h-BN on a SiO$_2$/Si substrate without the Pt NP array under the same conditions (950 °C, CH$_4$/Ar flow = 5/50 sccm). The PL emission was not observed in the sample (as shown in the blue spectrum of Supplementary Fig. 3c), indicating that the PL results originated from the in-plane heterostructures of GQD and h-BN. Furthermore, to check the PL emission of bare GQD without h-BN, a sample was prepared on a CVD-grown graphene monolayer via the O$_2$ plasma etching process by using the Pt NP array as a pattern mask (Supplementary Fig. 4a)[17]. We can easily distinguish the GQD arrays and SiO$_2$ regions by SEM contrast (Supplementary Fig. 4b). The PL peak at 410 nm wavelength was not observed from the bare GQD (as shown in the red spectrum of Supplementary Fig. 4c). Lastly, nano-sized holes of h-BN in the GQD/h-BN heterostructures may be one of the origins of the PL (Supplementary Fig. 5). To check this possibility, a h-BN sheet with nano-sized holes was prepared by using hydrogen-etching of h-BN through the annealing process on Pt NPs in H$_2$ atmosphere[15,18]. The CVD-grown single-layer h-BN was transferred onto the Pt NPs/SiO$_2$ substrate and etched away on the Pt NPs by annealing at 700 °C in H$_2$ flow (10 sccm). Supplementary Fig. 5b shows an AFM image of a hydrogen-etched h-BN film with holes transferred onto a SiO$_2$ substrate. No PL emission was observed for this sample (as shown in Supplementary Fig. 5c).

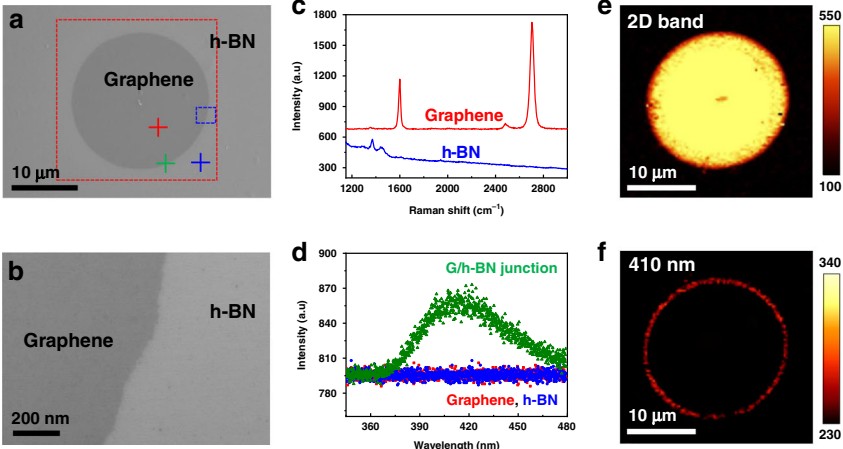

**Fig. 1 Characterization and PL analysis of graphene/h-BN in-plane heterostructures. a, b** SEM images of in-plane graphene/h-BN heterostructures fabricated by conversion reaction of h-BN on a Pt (111) single crystal. The location of the magnified SEM image, (**b**), is marked by the blue rectangle in **a**. **c** Raman spectra of graphene (red) and h-BN (blue) regions on a SiO₂/Si substrate using a 532-nm laser. **d** PL spectra of graphene (red), h-BN (blue) regions, and interface (green) using a 266-nm laser. **e, f** Raman and PL mapping images of the 2D band (2630–2730 cm⁻¹) and 410 nm wavelength, respectively, for the area inside the red square in **a**.

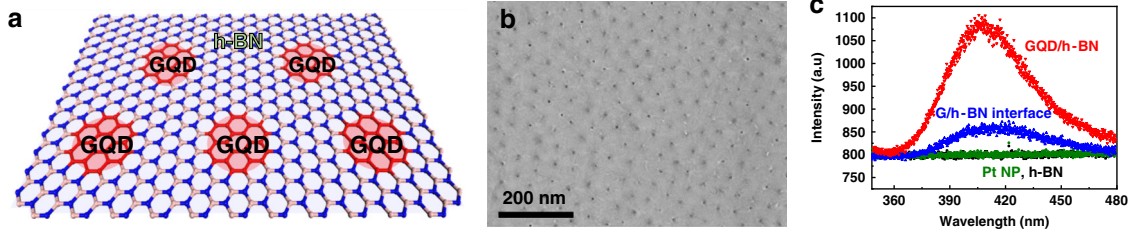

**Fig. 2 PL analysis of GQD/h-BN in-plane heterostructures. a** Schematic of the GQD/h-BN in-plane heterostructures fabricated by conversion of h-BN on Pt nanoparticles (Pt NPs). **b** SEM image of the GQD/h-BN heterostructures on a SiO₂/Si substrate. **c** PL spectra of GQD/h-BN (red), Pt NPs (green), and bare h-BN (back). The PL spectrum of G/h-BN interface (blue) is the same spectrum as green one in Fig. 1d. All of samples were measured on a SiO₂/Si substrate using 266-nm laser.

**Localized energy states at disordered G/h-BN boundary: origin of PL.** We confirmed that the PL emission is not due to the intrinsic properties in the GQDs and nano-sized holes of h-BN by the absence of PL in control experiments. Actually, it would be difficult to attribute the PL definitely to the 1D graphene/h-BN heterojunction because of the large area probed with a confocal Raman system. PL from surrounding areas is also measured. Focusing the laser only on the atomic-resolved interface is very challenging in the current PL technique. Based on this, it is difficult to conclusively attribute the observed PL emission to the 1D interface. Alternatively, to elucidate the origin of PL, we theoretically characterized the local energy states due to the disordered G/h-BN junction. First, we set-up an in-plane graphene/h-BN heterostructures lattice using molecular dynamics, in which there is an orientation mismatch between the graphene and h-BN like in a polycrystalline sample. Due to the mismatch, the G/h-BN junction (typically named grain boundary) is disordered; i.e., it is formed by a variety of non-hexagonal rings. Second, we set-up a tight-binding Hamiltonian to describe the electronic properties (see details in the Supplementary Information). We calculate the local density of states projected over all the sites at the interface by using the Kernel Polynomial Method[19] to evaluate the Green function; this shows two characteristic peaks at −1.2 eV and 2.0 eV (Fig. 3a). The energy separation of these peaks is, theoretically, characterized by a 390 nm wavelength, which is close to the experimental PL. The energy states corresponding to −1.2 eV and 2.0 eV are mainly located at the disordered G/h-BN heterojunctions, shown in Fig. 3b, c. PL is light emission after a

photoexcitation (photons excite electrons to a higher energy levels). During the photoexcitation and photoemission, there is a local charge redistribution along the interface which is only possible at the interface. The redistribution is allowed by the graphene charge carrier lifetime. Additionally, we also theoretically checked the size dependence using graphene/h-BN polycrystalline samples with three different average grain sizes: 10, 20, and 40 nm; as shown in Fig. 3a, the three samples show the same two peaks located at −1.2 eV and 2.0 eV, which indicates that there is no energy change of the localized state according to the graphene or h-BN domain size. Thus, we believe that localized energy states are formed at the disordered G/h-BN heterojunction, which result in PL emissions with a 400 nm wavelength. Our finding for the PL phenomenon at disordered G/h-BN heterojunction and its origin are different from the previously reported light emission in 2D materials caused by (i) doped Pt atoms[20–22], (ii) hydrogenated edges[23,24] in large nanoholes, (iii) atomic defects like $N_B V_N$ (nitrogen substitution at boron site + nitrogen vacancy)[25,26] and carbon substitution at nitrogen sites[27] in h-BN, and (iv) point defects in WSe₂[28–32]. Also, note that a slight difference of emission wavelength between theoretical and experimental results is due to limitations of a tight-binding Hamiltonian (TBH) calculation that cannot cover all experimental parameters such as electron-electron interaction effects or Coulomb screening caused by the generated excitons. To support above localized energy states at G/h-BN junctions, we need to show atomic structures at the disordered G/h-BN junctions. Thus, we measured annular dark-field scanning tunneling

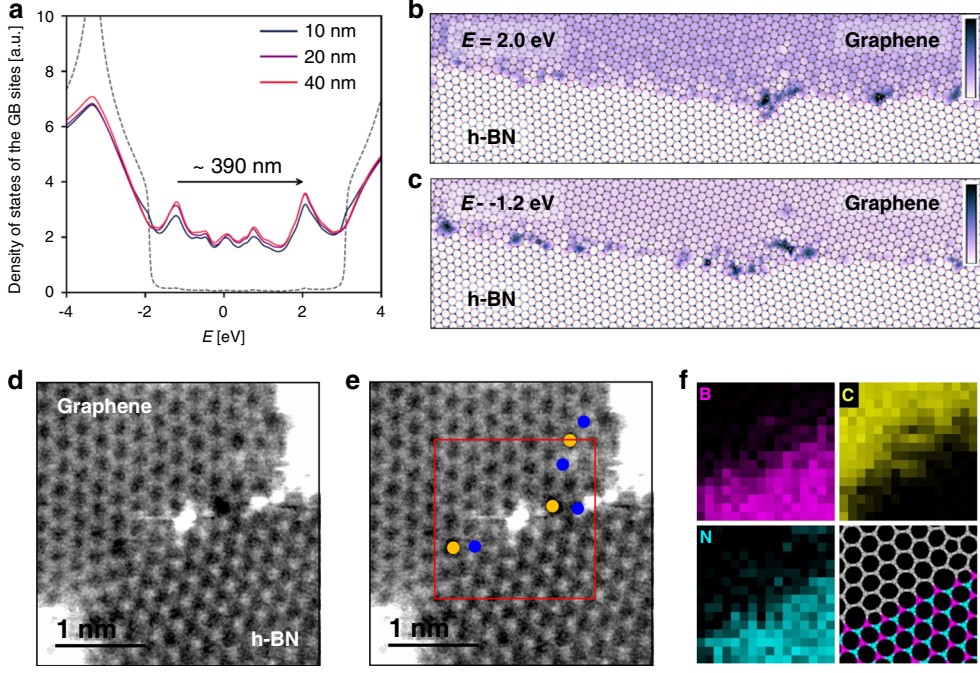

**Fig. 3 Disordered G/h-BN junctions and their electronic properties. a** Local density of state projected over all the sites at the graphene/h-BN interface for three different polycrystalline samples with 10 nm, 20 nm and 40 nm average grain size. As reference, dotted line corresponds to the density of states of a polycrystalline h-BN sample to show the energy location of the band gap. **b, c** Local density of states of each site in a disordered interface at 2.0 eV and −1.2 eV, respectively. It is clear that local density of states is higher at the structural defects of the interface. The units used are arbitrary. **d, e** Annular dark-field (ADF) STEM image showing a boundary between graphene and h-BN domains (top and bottom, respectively). Blue and orange circles denote 5- and 7-membered rings, respectively. The bright impurity atom found in the red box of **e** is identified as silicon by EELS, probably coming from the sample transfer process. **f** STEM-EELS elemental maps of boron, carbon, nitrogen and a possible structure model for the area indicated by a red square in **e**.

electron microcopy (ADF-STEM) images and electron energy loss spectroscopy (EELS) maps at the boundary of h-BN and graphene as shown in Fig. 3d–f and Supplementary Fig. S6. The high-resolution TEM images (Fig. 3d) clearly show the atomic structure of the interfaces between graphene (top region) and h-BN (bottom region), where we observed the disordered boundaries with 5-(blue spheres), and 7-membered rings (yellow spheres) consisting of C, B, and N atoms. Therefore, based on the TEM results to support our simulated model, we suppose that the blue emission comes from localized energy states at the disordered interface region between graphene and h-BN.

**Vertically stacked GQD/h-BN layers**. To improve the weak PL intensity from G/h-BN junctions, we fabricated the van der Waals heterostructures by vertically stacking GQD/h-BN layers using a layer-by-layer assembly with a wet-transfer method to increase the amount of G/h-BN junctions (Fig. 4a). Two types of stacked samples were prepared: one consists of four layers stack of GQD/h-BN films (4L GQD/h-BN) and the other inserts an additional h-BN intercalation layer between each GQD/h-BN film (4L GQD/h-BN with h-BN). As shown in Fig. 4b, the PL intensity of the 4L GQD/h-BN (red) was reduced compared to that of single-layer GQD/h-BN film (black). This indicates that the PL quenching still occurs in a structure that simply stacks GQD/h-BN films due to photon reabsorption and nonradiative energy transfer between GQDs in the vertical direction[33,34]. However, utilizing h-BN intercalation layers substantially improved the PL intensity since the charges generated on the GQD/h-BN film were not transported to GQDs in adjacent layers, blocked by the h-BN charge barrier (schematically shown in Fig. 4c). Although the intensity of 4L GQD/h-BN with h-BN does not improve to 4 times that of the single-layer GQD/h-BN due to the reabsorption problem (see details in the Supplementary Fig. S7), it is worth

noting that we realized G/h-BN junctions with the enhanced emission, isolated by h-BN intercalation layers in vertical directions. It indicates that the disordered G/h-BN junctions can be a potential component in optoelectronics. As an example, we fabricated a blue light emitting device using the GQD/h-BN heterostructures with the disordered G/h-BN junctions. With only GQD/h-BN heterostructures, it was confirmed that blue electroluminescence could be realized in LEDs without any energy donor in the emitting layer (see details in the Supplementary Figs. S8–S11).

## Discussion

In summary, we demonstrated blue light-emission at heterojunctions between graphene and h-BN domains. The emission likely originates from the localized energy states at the interface between graphene and h-BN on the in-plane heterostructures. We fabricated GQD/h-BN heterostructures and stacked them with h-BN intercalation layers to increase the total interface length, thereby improving PL emission. The stacked 3D heterostructures had the GQD structures isolated in the vertical and planar directions by the h-BN matrix, which improved their emission efficiency. This work suggests that the narrow 1D emission at heterojunctions of in-plane graphene/h-BN heterostructures could be used for future optoelectronic devices.

## Methods

**Growth of simple in-plane graphene/h-BN heterostructures**. The in-plane heterostructure of graphene and h-BN was prepared by a catalytic conversion reaction of h-BN to graphene on a Pt metal substrate[17]. A single layer of h-BN was synthesized on a Pt (111) single crystal using ammonia borane as a precursor by CVD method[35]. The resulting complex, h-BN/Pt(111), was then loaded into the center of a vacuum quartz tube placed in a furnace. The tube was pumped down to 0.21 torr with pure argon gas (50 sccm), and then heated to 1000 °C at a steady rate for over 40 min. The reaction was initiated with a flow of methane (5 sccm) and

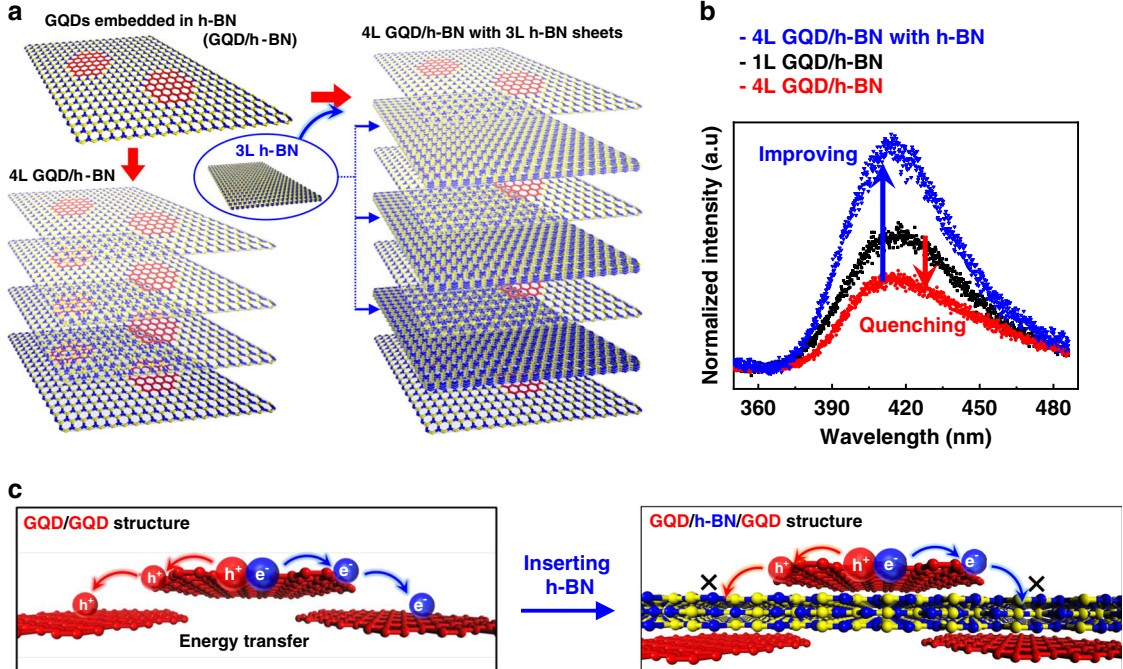

**Fig. 4 Vertically stacked GQD/h-BN layers with h-BN barriers. a** Schematic of the preparation process for two kinds of stacked GQD/h-BN films. **b** PL spectra of stacked GQD/h-BN on a SiO₂/Si substrate: a four layer stack of GQD/h-BN films (4L GQD/h-BN, red spectrum), 4L GQD/h-BN with an additional h-BN intercalation layer between each GQD/h-BN film (4L GQD/h-BN with h-BN, blue spectrum), and a single layer of GQD/h-BN film (black spectrum, same as the red spectrum in Fig. 2d). **c** Schematic of the nonradiative charge transfer between GQDs in adjacent layers.

argon (50 sccm) gases to produce the in-plane heterostructures with a shorter 10 min reaction time.

**Growth of graphene quantum dots embedded in a h-BN sheet.** A Pt nanoparticle (NP) array on a SiO₂ substrate was prepared using self-patterning diblock copolymer micelles[36]. A single-layer of polystyrene-block-poly(4-vinylpyridine) (PS-P4VP) micelles with H₂PtCl₆, a precursor of Pt NPs, in their cores was spin-coated on the SiO₂ substrate. To fabricate the Pt NP array, the micellar film on the SiO₂ was annealed at 400 °C for 30 min in air. The h-BN grown on Pt substrate were transferred onto the Pt NPs/SiO₂ substrate using a wet-transfer method (electrochemical delamination). Then, the sample was loaded into the center of a vacuum quartz tube in a furnace for the conversion reaction. The tube was pumped down to 0.21 Torr with pure Argon gas (50 sccm). Then the furnace was heated to 950 °C in 40 min. When the reaction started, methane gas (5 sccm) with Argon (50 sccm) was flown as the source for graphene growth. During the reaction, the h-BN region on the Pt NPs was converted to graphene, and after 10 min of growth a uniform GQD array embedded in the h-BN film was obtained[15].

**Transfer of GQD/h-BN films to other substrates.** The GQD/h-BN film on Pt NPs/SiO₂ was transferred onto other substrates via a wet-transfer method using HF and an aqua regia solution. First, polystyrene (PS) was spin-coated on the sample, and it was immersed in a HF solution (5% in DI water) to remove the SiO₂ layer. Then, the floating PS film was transferred to the aqua regia solution (3:1 mixture of hydrochloric acid and nitric acid) to remove the Pt NPs. Finally, the film was transferred onto the desired substrate and the PS film was removed with toluene to obtain a GQD/h-BN film on the substrate.

**Characterization.** Scanning electron microscopy (Verios 460, FEI) and atomic force microscopy (Dimension Icon, Bruker) were used to determine the surface morphology of the samples. PL and Raman spectra were measured using a micro Raman spectroscope (alpha 300, WITec GmbH) with 266-nm and 532-nm lasers, respectively. The h-BN/graphene heterostructure sample was transferred to a microporous silicon nitride membrane TEM grid (Alliance Biosystems, Inc.) for scanning transmission electron microscopy (STEM) combined with electron energy loss spectroscopy (EELS). JEOL JEM-2100F electron microscope equipped with JEOL Delta spherical aberration correctors and Gatan Quantum electron spectrometer was operated at an electron accelerating voltage of 60 kV. EELS chemical maps of boron, carbon and nitrogen were obtained by measuring their K edge signals at each point of a scanned area.

**Fabrication of GQD/h-BN-based LED device and evaluation.** A GQD-based LED device consists of the following layers (of corresponding thickness): indium

tin oxide (ITO) (150 nm), poly(3,4-ethylenedioxythiopehe):poly(styrene sulfonate) (PEDOT:PSS) (40 nm), poly(N-vinylcarbazole) (PVK) (20 nm), an emitting layer, 2,2′,2″-(1,3,5-benzinetriyl)-tris(1-phenyl-1-H-benzimidazole) (TPBI) (60 nm), lithium fluoride (LiF) (1 nm), and Al (100 nm). A glass substrate pre-deposited with a 150 nm thick ITO electrode (half-etched) was cleaned in an ultrasonic bath using de-ionized (DI) water, acetone, and isopropyl alcohol, and then it was treated with air plasma using a plasma cleaner (PDC-32G, Harrick Plasma). The PEDOT:PSS (CLEVIOS P VP AI 4083) was deposited by spin-coating (3000 rpm for 40 s) and annealed at 150 °C for 30 min. Then, the PVK (Mₙ: 25,000–50,000) was deposited onto the PEDOT:PSS layer by spin-coating (1500 rpm for 40 s) using a PVK solution (15 mg/mL in chlorobenzene) and annealed at 120 °C for 20 min. The fabricated GQD/h-BN was transferred onto the top of the PVK by using the wet-transfer method. The samples with PVK or GQD/h-BN were loaded into a thermal evaporator (HS-1100, Digital Optics & Vacuum) for deposition of the TPBI and LiF/Al under high vacuum (~10⁻⁶ Torr). The current density-luminance-voltage characteristics of the fabricated LED devices were evaluated using a Keithley 2400 source meter and a calibrated photodiode (FDS100, Thorlab). The EL spectra were measured by a fiber-optic spectrometer (EPP2000, StellarNet). All measurement processes were performed in an inert environment.

## Data availability
All data that support the findings of this study are present in the paper and the supplementary materials, and additional data are available from the corresponding author upon reasonable request.

## Code availability
The codes that are used to generate results in this paper are available from the corresponding author upon reasonable request.

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

## Acknowledgements

This work was supported by the research funds (NRF-2017R1E1A1A01074493 and NRF-2019R1A4A1027934) and the grant (CASE-2013M3A6A5073173) from the centre for Advanced Soft Electronics under the Global Frontier Research Program through the National Research Foundation by the Ministry of Science and ICT, Korea. H.S.S. and K.S. thank the A3 foresight program for their collaboration. Y.S., Y.-C.L. and K.S. acknowledge JSPS KAKENHI Grant Numbers JP19K05223, JP18K14119 and JP16H06333, respectively. S.R. acknowledges the European Union's Horizon 2020 research and innovation programme under Grant Agreement No. 881603 (Graphene Flagship). ICN2 is funded by the CERCA Programme/Generalitat de Catalunya, and is supported by the Severo Ochoa program from Spanish MINECO (Grant No. SEV-2017-0706). J.E.B.-V. acknowledges funding from PAIP Facultad de Química, UNAM (Grant No. 5000-9173).

## Author contributions

H.S.S. planned and supervised this project. G.K. and M.K. contributed to the growth of in-plane heterostructures. J.J. and B.H.S. provided the Pt nanoparticles array on SiO₂ substrates. G.K and K.Y.M. performed the characterizations (PL, Raman, SEM) and data analyses. G.K., M.P., J.S., S.Y., and S.J. contributed fabrication of GQD/h-BN-based LED device and evaluation. J.E.B-V. and S.R. designed and performed the numerical calculation and the modelling data summary. Y.S., Y-C.L., and K.S. performed the TEM analysis on the in-plane graphene/h-BN heterostructures. All authors discussed the results and commented on the paper.

## Competing interests

The authors declare no competing interests.
