## [Peer Review File · Nature Communications]

Reviewers' comments:

Reviewer #1 (Remarks to the Author):

The authors present an interesting study on graphene quantum dots (GQD)/h-BN one-dimensional heterostructure for a light-emitting material. The key results here is that the blue light-emitting from graphene/h-BN boundaries is investigated by both PL experiment and theoretical calculations. In addition, the LED device utilizing the GQD/h-BN junction array as the emitting layer is demonstrated. The results here provide a valuable path forward for other researchers in the field. However, there are a few issues that should be addressed before publication.

1.The authors insist that the origin of the light emitting from GQD/h-BN junction is the two localized states at the disorders of the GQD/h-BN grain boundaries. The authors should address which structure of the disorders (e.g., 5-membered ring or 7-membered ring, and combination of C, B, and N atoms, etc.) at the GQD/h-BN grain boundaries attributed the forming of the two localized states. In addition, the authors should show the clear evidence of the defect structures at the grain boundary contributing to the light emitting from the GQD/h-BN junction, such as STM image or TEM image of the grain boundary in atomic scale.

2.The wavelength of the PL peak from the GQD/h-BN junction is 410 nm in the experiment. However, the theoretically predicted light emitting wavelength from two localized state at the GQD/h-BN grain boundary is 390 nm. The author should the reasons for difference of the emitting wavelength between the experiment and theory.

3.The authors described the external quantum efficiency (EQE) of the LED device using the GQD/h-BN junction. There is, however, no information on the quantum yield of the GQD/h-BN junction in the PL experiments. This is the valuable information on the characteristic of the luminescence materials. The authors should address the quantum yield of the GQD/h-BN junction in the PL experiment.

4.The PL intensity of 4 layers of GQD/h-BN with h-BN did not improve to 4 times that of the single layer GQD/h-BN. The authors insist that the reason of low PL intensity is due to the reabsorption problem. This is too unclear for the readers. The authors should address detailed physically reasonable reasons for it.

5.The EQE of the LED device using GQD/h-BN layer is 0.5 %. The authors insist that the further optimization of the fabrication process improves the performance of the device. The authors should address the theoretical maximum EQE of the GQD/h-BN layer and the more concrete plan to improve the device performance.

6.For reader readability and ease of understanding, the authors should show the optical image of the GQD/h-BN structures and the LED device with the GQD/h-BN layer.

7.The authors should show the reproducibility of the experiment both the PL experiments and the device performance of the LED with the GQD/h-BN layer.

8.In Page 11, Line 226, "Fig.A" is probably mere typo. Please correct it.

Reviewer #2 (Remarks to the Author):

The paper by Gwangwoo Kim and coworkers reports blue light emission from hBN-graphene in-plane interfaces, with the graphene being obtained by controlled conversion of hBN into graphene on a Pt catalytic substrate, and such light emission is attributed to optically active edge states formed at the disordered interface. Then, multiple layers of graphene quantum dots embedded in hBN are stacked on top of each other and combined in LEDs, to show electroluminescence with an external quantum efficiency around 0.5%.

The results may be interesting, but personally I found the article lacking focus, poorly structured and difficult to follow, and as such the authors could have done a much better job at conveying their message (and could have shown a bit more mercy for the reviewers). I would like to see some major restructuring before deciding whether to recommend it or not for publication.

To start getting into the details, the paper comprises 2 main sections: on the one side the report of new light emitters, on the other side the realization of LEDs with such emitters, but right now both are underwhelmingly presented and lack a strong motivation, essential information and sufficient detail. The authors should consider whether it may be better to split the paper in 2, and devote 1 full paper to each topic. Anyway, a thorough revision is required.

-Concerning the fabrication of the emitters:

1) the finding may be interesting, but I feel that it lacks a strong motivation. In the past few years, light emission from defects has been reported in a variety of 2D and layered materials, often correlated with single-photon signatures, deterministic positioning and electroluminescence (it may help to refer to the works of the groups of Igor Aharonovich & Milos Toth, Dirk Englund, Jon Finley & Alex Holleitner, etc. for hBN and Rudolf Bratschitsch, Atac Imamoglu, Marek Potemski, Mete Atatüre & Andrea Ferrari, Brian Gerardot, etc. on TMDs), so the authors are not sailing uncharted waters and may well want to underline better the relevance of using a graphene/hBN system, starting with writing a more comprehensive introduction.

2) There's very limited insight on the optical properties of the emitters. Right now it's very difficult to tell whether they are promising or not. Can the authors add/specify more in terms of, for instance, counts per second, quantum efficiency, linewidth, and comment on the stability of such properties across the sample and in time?

3) Some essential information on the fabrication of each sample must be included in the main text before the experimental investigation is described. As it is, one needs to constantly jump to the methods to understand which sample is being investigated, under which conditions it was made, etc. Given the variety of the fabrication processes considered, and the often nuanced differences among them, the narrative flow and the logical reasoning is severely hampered.

Moreover, fabrication details must be added everywhere. As an example, the authors use O₂ plasma to etch graphene dots, but a proper description of the process is completely missing.

4) The authors perform a very thorough set of different experiments to find the origin of their light emission. Essentially they test a sample with a large (many microns) graphene circle embedded in hBN, and an analogous design but with much smaller graphene QDs embedded in hBN. Both designs produce light emission, with the former clearly showing that the emission is coming from the edge. To exclude alternative options, they also produce 2 control samples, each with one independent phase, i.e. 1 with isolated graphene QDs and 1 with hBN full of holes, none of which is showing emission, thus they use this evidence, plus the decent agreement with calculations, as evidence that the light emission originates from disordered interface states forming at the G/hBN junction. This could very well be the case. However, digging a bit (not easy!) among the fabrication details, I understood that while graphene macrodots in hBN, graphene QDs in hBN and graphene control dots on SiO₂ are all processed at similar temperature (950 or 1000 C), the hBN control sample with holes is processed at 700 C. It is very well known by now that hBN is full of different optically active atomic impurities, from UV to NIR (look at these for a quick overview: Aharonovich, I., Englund, D. & Toth, M. Solid-state single-photon emitters. *Nature Photon* 10, 631–641 (2016), Atatüre, M., Englund, D., Vamivakas, N. et al. Material platforms for spin-based photonic quantum technologies. *Nat Rev Mater* 3, 38–51 (2018).), and annealing plays a role in their activation, so what may be happening is that impurities in hBN have a high enough diffusion coefficient at 1000 C to segregate in large quantities at the edges of the crystal, thus also at the

graphene/hBN interface, while not having such a mobility at 700 C to be seen by PL. The authors should thus consider if this hypothesis is realistic, what would be the temperature dependence of the diffusion of impurities in hBN between 700 and 1000 C, look for signs of luminescence at the external edges of the hBN/graphene samples, and if those are dark, ideally processing an hBN control sample with holes in the exact same conditions of the graphene and graphene/hBN samples, and checking for signs of luminescence there.

-Concerning the LED

5) As with the previous part, also the LED portion needs a stronger motivation. I find the sentence: "We believe this is a significant achievement in that we have fabricated a GQD-self-emitting device using a single-layer GQD structure isolated by h-BN matrix." quite tautological. We can rephrase it as: "We have fabricated this new structure and we believe it to be a significant achievement". This is a belief, not a motivation.

The field is crammed with a plethora of similar works using many different 2D materials-based hybrid structures to make LEDs, the most recent also adding more advanced features such as cavity integration, mechanical flexibility, semi-transparency, large-area etc. (to point at the general concept, one early example is Withers, F., Del Pozo-Zamudio, O., Mishchenko, A. et al. Light-emitting diodes by band-structure engineering in van der Waals heterostructures. Nature Mater 14, 301–306 (2015), which however reports an incorrect EQE by a few orders of magnitude -lamentably never corrected- so I would not use it as a benchmark for the EQE specifically). Similar to the case of the emitters, the introduction does little to position the LEDs within the wider context of the current state-of-the-art, with the only exception of a reference to colloidal GQD-based LEDs, a quite different system, where GQDs are the optically active component. To go to the point: what does set this work clearly apart from the rest? This is not very obvious to me.

6) the EQE is not stellar compared to commercial and state-of-the-art LEDs, but quite in-line with reports from analogous systems. However, I would like to see a vision from the authors. What is their outlook regarding their platform? What is their aim? Where do they see it going? This is completely missing from the conclusions, which right now are just summarizing their findings. A good vision would help strengthen the motivation.

7) Can the authors explain in detail how they calculate the external quantum efficiency and which parameters they use?

All the Best

Reviewer #3 (Remarks to the Author):

This paper is exploring the PL and EL from regions at the interface between graphene and hBN. The authors claim of PL localization in the blue spectral region by the graphene/hBN interface. The ability to obtain PL from 2D materials is an important area, but this paper lacks substantial structural information to draw conclusion. At the moment the authors cannot be sure what is the source of the PL, and can only speculate.

1. The authors have to perform atomic resolution TEM imaging of the grain boundary region between the graphene and hBN to determine the actual structure.

2. there is no actual evidence that graphene quantum dots have been produced. The evidence is missing. The authors have to perform some annular dark field scanning transmission electron microscopy to show the carbon regions within the hBN host. Or EELs mapping to show C segregated in the hBN. It could be that there is diffusion of C, B, N in these regions

Overall, without atomic structure details, the paper cannot be published in a high impact journal because the authors cannot know the cause of the PL signals. Without knowing the cause (i.e structure-property correlation), the paper lacks sufficient depth and novelty for publication.

Point by Point Replies to Reviewer's Comments

Reviewer #1

The authors present an interesting study on graphene quantum dots (GQD)/h-BN one-dimensional heterostructure for a light-emitting material. The key results here is that the blue light-emitting from graphene/h-BN boundaries is investigated by both PL experiment and theoretical calculations. In addition, the LED device utilizing the GQD/h-BN junction array as the emitting layer is demonstrated. The results here provide a valuable path forward for other researchers in the field. However, there are a few issues that should be addressed before publication.

Reply: We appreciate Reviewer #1's positive comments.

1. The authors insist that the origin of the light emitting from GQD/h-BN junction is the two localized states at the disorders of the GQD/h-BN grain boundaries. The authors should address which structure of the disorders (e.g., 5-membered ring or 7-membered ring, and combination of C, B, and N atoms, etc.) at the GQD/h-BN grain boundaries attributed the forming of the two localized states. In addition, the authors should show the clear evidence of the defect structures at the grain boundary contributing to the light emitting from the GQD/h-BN junction, such as STM image or TEM image of the grain boundary in atomic scale.

Reply: We could not identify any particular disordered structure at the boundary which contributes to the two localized states. Based on our results, however, we think that the two localized states are due to combination of 5- and 7-membered rings including C, B, and N atoms at the boundary. Our calculation results showed localized energy states around the boundaries with 5- and 7-membered rings consisting of C, B, and N atoms. The atomic structures at the boundary from the calculation results are well supported by TEM results below (Figures R1 and R2).

We measured annular dark-field scanning tunneling electron microscopy (ADF-STEM) images and electron energy loss spectroscopy (EELS) maps at the boundary of hBN and graphene as shown in Figures R1 and R2. The results clearly show the atomic structures at the boundaries of hBN/graphene in-plane heterostructures, where we observed the disordered boundaries with 5-, and 7-membered rings consisting of C, B, and N atoms. It is consistent with our calculation results. Thus, theoretical and experimental results prove the two localized energy states which can be the origin of PL at the boundary.

We add the sentences and TEM results in the manuscript (8 page, Fig. 3d-f) and supplementary information (Supplementary Fig. S6).

“To support above localized energy states at G/BN junctions, we need to show atomic structures at the disordered G/BN junctions. Thus, we measured annular dark-field scanning tunneling electron microscopy (ADF-STEM) images and electron energy loss spectroscopy (EELS) maps at the boundary of h-BN and graphene as shown in Fig. 3d-f and Supplementary Fig. S6. The high-resolution TEM images (Fig. 3d) clearly show the atomic structure of the interfaces between graphene (top region) and h-BN (bottom region), where we observed the disordered boundaries with 5- (blue spheres), and 7-membered rings (yellow spheres) consisting of C, B, and N atoms. Therefore, we suppose that the emission comes from new localized energy states in the disordered interface region between the graphene and

h-BN.”

Figure R2. STEM-EELS analysis of h-BN/graphene heterostructure. a. Annular dark-field (ADF) STEM image showing a boundary between h-BN and graphene domains (left and right, respectively). b. Magnified ADF-STEM image of the area indicated by a green square in (a). Blue and orange spheres denote 5- and 7-membered rings, respectively. c. EELS data acquired at the positions labeled as #1 and #2 in (a). d. STEM-EELS elemental maps of boron, carbon, nitrogen and a possible structure model for the area indicated by a red square in (a).

2. The wavelength of the PL peak from the GQD/h-BN junction is 410 nm in the experiment. However, the theoretically predicted light emitting wavelength from two localized state at the GQD/h-BN grain boundary is 390 nm. The author should the reasons for difference of the emitting wavelength between the experiment and theory.

Reply: We used a tight-binding Hamiltonian (TBH) with parameters obtained from DFT calculations. These parameters were fitted to reproduce the polarization of the interface; thus, the charge accumulation at the interface was taken into account as an effective electric potential. Our TBH calculations are however not treating exactly the electron-electron interaction, which is out of reach for disordered systems. Thus, we cannot expect a perfect agreement with the experimental data. Furthermore, owing to some Coulomb screening caused by the generated excitons, a difference between theoretical and experimental emitting wavelength is also expected. In fact, the obtained difference (20 nm, or equivalent ~ 150 meV) is satisfactory given that electron-electron interaction effects are expected to play a role in the PL at the interface.

We add the sentences in the manuscript (8 page).

“Also, note that a slight difference of emitting wavelength between theoretical and experimental results is due to limitations of tight-binding Hamiltonian (TBH) calculation that cannot cover all experimental parameters such as electron-electron interaction effects or Coulomb screening caused by the generated excitons.”

3. The authors described the external quantum efficiency (EQE) of the LED device using the GQD/h-BN junction. There is, however, no information on the quantum yield of the GQD/h-BN junction in the PL experiments. This is the valuable information on the characteristic of the luminescence materials. The authors should address the quantum yield of the GQD/h-BN junction in the PL experiment.

Reply: We agree that the information of the luminescence characteristics such as the quantum yield (QY) is important. Unfortunately, however, we failed in obtaining QY of the GQDs/h-BN sample when we tried it by using a commercially available instrument with a monochromator and Xe lamp (FP-8500ST from JASCO). This would be due to very low absorption (Fig. R3) and PL. In fact, PL of the sample was relatively low and measured by micro-Raman system with 266 nm laser. We do not have an instrument to measure QY with a laser. Furthermore, we want to focus on new, unprecedented PL phenomenon and its origin in this revised manuscript. Thus, we moved the LED device section to Supplementary Information which is just an example of potential applications.

Figure R3. UV/visible absorption spectra of h-BN and GQDs/h-BN layers.

4. The PL intensity of 4 layers of GQD/h-BN with h-BN did not improve to 4 times that of the single layer GQD/h-BN. The authors insist that the reason of low PL intensity is due to the reabsorption problem. This is too unclear for the readers. The authors should address detailed physically reasonable reasons for it.

Reply: It is known that the stacks of GQDs in the solid state induced photon reabsorption and nonradiative energy transfer which indicates partial PL quenching (*J. Phys. Chem. C* **2016**, *120*, 29432; *RSC Adv.* **2015**, *5*, 57425). That is, at the stacked sample, the excited electrons of GQDs may nonradiatively relax to ground states through couplings with neighboring ones (by reabsorption and energy transfer). We solved the issue of nonradiative energy transfer by inserting 3L h-BN as an intercalation layer between GQD/h-BN layers, but the generated photons from GQD/h-BN can be still re-absorbed by adjacent layers. However, note that 3L h-BN as an intercalation layer may not fully prevent the nonradiative energy transfer because charge tunneling through 3L h-BN may occur still.

Comparing the PL emission in 1 layer-, 2 layers-, 3 layers-, and 4 layers-stacked GQD/h-BN films with h-BN intercalation layers, the extent of enhanced PL intensity (blue solid line) did not increase by 2, 3, or 4 times (red dotted line) proportionally as shown in Figure R4.

Figure R4. Comparison of PL intensity of the different numbers of GQD/h-BN layers with 3L h-BN as an intercalation layer.

We add above result in the supplementary information (Supplementary Fig. S7).

5. The EQE of the LED device using GQD/h-BN layer is 0.5 %. The authors insist that the further optimization of the fabrication process improves the performance of the device. The authors should address the theoretical maximum EQE of the GQD/h-BN layer and the more concrete plan to improve the device performance.

Reply: The theoretical EQE is the product of the internal quantum efficiency (IQE) and light out-coupling factor (η_{out}), $EQE=IQE \times \eta_{out}$. IQE is obtained from $IQE=\beta \times \gamma \times \Phi_{PL}$, where β is the exciton generation factor resulting in photons, γ is the carrier balance ratio of holes and electrons, and Φ_{PL} is the photoluminescence (PL) quantum yield (PLQY). As we mentioned above, however, we have not been able to obtain the value of PLQY due to the limitation of our equipment. Thus, we cannot calculate the maximum EQE. Please note that this work focuses on unprecedented PL and its origin at the disordered boundary of h-BN and graphene, rather than the LED device. We moved the LED device as a just potential application to Supplementary Information, and thus will do further study for optimization and in-depth characterization of the LED device.

In addition, we will have future plan to improve the EQE value of the LED devices. Increasing the amounts of G/h-BN heterojunctions would be the first way to improve the performance. The stacking process mentioned in the paper (Fig. 4) is also one way to increase the amounts. Also, we can try the increased density of GQDs by increasing the density of Pt particles during the fabrication of GQDs/h-BN in-plane heterostructures. Lastly, in terms of LED device structure, it is an important method to select electron and hole transport layers matched with the band alignment of the heterojunctions.

6. For reader readability and ease of understanding, the authors should show the optical image of the GQD/h-BN structures and the LED device with the GQD/h-BN layer.

Reply: Please note that the average size of GQDs in h-BN matrix is 7 nm and thus it is not possible to show in an optical image or optical microscope (OM) image. Instead, we showed a SEM image of the GQDs array in h-BN matrix in Figure 2b. The detailed analysis on the GQD/h-BN structure was demonstrated in our previous report (*Nat. Commun.* **2019**, *10*, 230).

We will add an optical image of the LED devices with the GQD/h-BN layer in the inset of Supplementary Fig. 8d, and an image for a detailed structure of the LED device in Figure R5 which will be added in Supplementary Information (Supplementary Fig. S9).

Supplementary Fig. 8d. (d) EL spectrum for the fabricated GQD/h-BN devices. Inset is the optical image of the LED device with the GQD/h-BN layers. The GQD/h-BN region is marked with a red dotted line.

Figure R5. The optical images of the detailed LED device with the GQD/h-BN layers. **a**, A photograph of organic materials-based LED devices with GQD/h-BN films. The GQD/h-BN region is indicated by a yellow dotted line. **b**, Scheme of cross-sectional device structure marked by orange line in Figure R4a. **c-e**, Enlarged optical microscopic images of each position. (b, red box; c, blue box; d, green box). Scale bars are 100 μm .

7. The authors should show the reproducibility of the experiment both the PL experiments and the device performance of the LED with the GQD/h-BN layer.

Reply: We show reproducibility of five GQD/h-BN samples for PL and four LED devices with GQD/h-BN layers in Figures R6 and R7 below. The mean and standard deviation of PL intensity and peak position were 299.2 ± 32.34 and 409.16 ± 1.94 nm, respectively. The mean and standard deviation of EL luminance and peak position were 20.75 ± 6.33 and 407.35 ± 0.76 nm, respectively. We add above results in the supplementary information (Supplementary Fig. S2 and S11).

Figure R6. Comparison of PL intensity and wavelength on the different five GQD/h-BN samples. The mean and standard deviation of PL intensity and peak position were 299.2 ± 32.34 and 409.16 ± 1.94 nm, respectively.

Figure R7. Comparison of EL luminance and wavelength on the different four devices with GQD/h-BN layers. The mean and standard deviation of EL luminance and peak position were 20.75 ± 6.33 and 407.35 ± 0.76 nm, respectively.

8. In Page 11, Line 226, “Fig.A” is probably mere typo. Please correct it.

Reply: Thank you for pointing it out. We correct it.

Reviewer #2

The paper by Gwangwoo Kim and coworkers reports blue light emission from hBN-graphene in-plane interfaces, with the graphene being obtained by controlled conversion of hBN into graphene on a Pt catalytic substrate, and such light emission is attributed to optically active edge states formed at the disordered interface. Then, multiple layers of graphene quantum dots embedded in hBN are stacked on top of each other and combined in LEDs, to show electroluminescence with an external quantum efficiency around 0.5%.

The results may be interesting, but personally I found the article lacking focus, poorly structured and difficult to follow, and as such the authors could have done a much better job at conveying their message (and could have shown a bit more mercy for the reviewers). I would like to see some major restructuring before deciding whether to recommend it or not for publication.

To start getting into the details, the paper comprises 2 main sections: on the one side the report of new light emitters, on the other side the realization of LEDs with such emitters, but right now both are underwhelmingly presented and lack a strong motivation, essential information and sufficient detail. The authors should consider whether it may be better to split the paper in 2, and devote 1 full paper to each topic. Anyway, a thorough revision is required.

Reply: We thank Reviewer #2 very much for his/her suggestion on restructuring of our paper. In this revised manuscript, we want to highlight new, unprecedented luminescence property at the disordered boundary of h-BN and graphene, which is the first report in this field. In this revised manuscript, we add actual atomic structures at the boundary by TEM, supporting the disordered boundary that we proposed by the calculation results for localized energy states as an origin of the new PL around 400 nm. Thus, we believe that we now focus on new PL phenomena with strong evidences in this revised manuscript. To avoid any confusion due to two main sections (new PL phenomenon and LED device) that Reviewer #2 pointed out, we deleted the LED device section in the main text. However, we think that we cannot prepare a separate full paper on the LED device part because its efficiency is low at the moment. Thus, we want just to show it as an example that it can be used in optoelectronics in future and moved it to Supplementary Information.

-Concerning the fabrication of the emitters:

1) The finding may be interesting, but I feel that it lacks a strong motivation. In the past few years, light emission from defects has been reported in a variety of 2D and layered materials, often correlated with single-photon signatures, deterministic positioning and electroluminescence (it may help to refer to the works of the groups of Igor Aharonovic & Milos Toth, Dirk Englund, Jon Finley & Alex Holleitner, etc. for hBN and Rudolf Bratschitsch, Atac Imamoglu, Marek Potemski, Mete Atatüre & Andrea Ferrari, Brian Gerardot, etc. on TMDs), so the authors are not sailing uncharted waters and may well want

to underline better the relevance of using a graphene/hBN system, starting with writing a more comprehensive introduction.

Reply: We thank the Reviewer #2 for letting us know the research groups. They reported single photon emission at point defects ($N_B V_N$: nitrogen substitution at boron site + nitrogen vacancy^{20, 21} & carbon substitution at nitrogen sites²²) in h-BN and point defects in WSe_2 ²³⁻²⁵. To date, however, there is no report on PL at disordered boundary of h-BN and graphene. We demonstrate this new PL at such sharp heterojunctions as a very fundamental study. Furthermore, we proved the disordered boundary consisting of 5- and 7-membered rings with combinations of C, B, and N atoms as the PL origin by theoretical and TEM results.

We add the sentence in the page 8 of the revised manuscript.

“Our finding for the unprecedented PL phenomenon at disordered G/BN heterojunction and its origin are different from the previously reported light emission in 2D materials caused by atomic defects like $N_B V_N$ (nitrogen substitution at boron site + nitrogen vacancy)^{20, 21} and carbon substitution at nitrogen sites²² in h-BN, and point defects in WSe_2 ²³⁻²⁵.”

20. Tran, T. et al. Quantum Emission from Hexagonal Boron Nitride Monolayers. *Nat. Nanotechnol.* **11**, 37–41 (2016).
21. Grosso, G. et al. Tunable and High-Purity Room Temperature Single-Photon Emission from Atomic Defects in Hexagonal Boron Nitride. *Nat. Commun.* **8**, 705 (2017).
22. Bourrellier, R. et al. Bright UV Single Photon Emission at Point Defects in h-BN. *Nano Lett.* **16**, 4317-4321 (2016).
23. He, Y. et al. Single Quantum Emitters in Monolayer Semiconductors. *Nat. Nanotechnol.* **10**, 497–502 (2015).
24. Koperski, M. et al. Single Photon Emitters in Exfoliated WSe_2 Structures. *Nat. Nanotechnol.* **10**, 503–506 (2015).
25. Chakraborty, C. et al. Voltage-Controlled Quantum Light from an Atomically Thin Semiconductor. *Nat. Nanotechnol.* **10**, 507–511 (2015).

2) There's very limited insight on the optical properties of the emitters. Right now it's very difficult to tell whether they are promising or not. Can the authors add/specify more in terms of, for instance, counts per second, quantum efficiency, linewidth, and comment on the stability of such properties across the sample and in time?

Reply: We show reproducibility of five GQD/h-BN samples for PL and four LED devices with GQD/h-BN layers in Figures R6 and R7 above (in the reply to Point 7 by Reviewer #1). We could not obtain quantum efficiency because of very low absorption and PL. This would be a future research topic.

3) Some essential information on the fabrication of each sample must be included in the main text before the experimental investigation is described. As it is, one needs to constantly jump to the methods to understand which sample is being investigated, under which conditions it was made, etc. Given the variety of the fabrication processes considered, and the often nuanced differences among them, the narrative flow and the logical reasoning is severely hampered.

Moreover, fabrication details must be added everywhere. As an example, the authors use O₂ plasma to etch graphene dots, but a proper description of the process is completely missing.

Reply: Thank you for the suggestions. We add essential information on the fabrication of each sample in the main text as follows, even though we described details in the supplementary information.

“In order to check PL emission on bare GQD without h-BN, the sample was prepared using a single-layer graphene grown on Pt foil and O₂ plasma etching process, shown in Figure S3a. First, the graphene monolayer grown on the Pt foil is transferred onto a SiO₂/Si substrate by electrochemical delamination method. After aligning the Pt NPs array on the top of graphene by self-assembly process (*Chem. Mater.* **2015**, *27*, 7003), O₂ plasma etching is performed on the sample at 50 W for 1 min, and in the graphene etching process, Pt NPs array is used as a pattern mask. After the reaction, Pt NPs are removed with aqua regia solution.”

“The h-BN sheet with nano-sized holes was prepared by using hydrogen-etching of h-BN through the annealing process on Pt NPs in H₂ atmosphere (*Nano Lett.* **2015**, *15*, 4769). First, after aligning the Pt NPs array on the SiO₂/Si substrate by self-assembly process, a single-layer h-BN CVD-grown on the Pt foil (*Nano Lett.* **2013**, *13*, 1834) was transferred onto the Pt NPs/SiO₂ substrate by electrochemical delamination method. And then, we carried out hydrogen-etching of h-BN on Pt NPs by annealing at 700 °C in H₂ flow (30 sccm) for 10 min.”

4) The authors perform a very thorough set of different experiments to find the origin of their light emission. Essentially they test a sample with a large (many microns) graphene circle embedded in hBN, and an analogous design but with much smaller graphene QDs embedded in hBN. Both designs produce light emission, with the former clearly showing that the emission is coming from the edge. To exclude alternative options, they also produce 2 control samples, each with one independent phase, i.e. 1 with isolated graphene QDs and 1 with hBN full of holes, none of which is showing emission, thus they use this evidence, plus the decent agreement with calculations, as evidence that the light emission originates from disordered interface states forming at the G/hBN junction. This could very well be the case.

However, digging a bit (not easy!) among the fabrication details, I understood that while graphene macrodots in hBN, graphene QDs in hBN and graphene control dots on SiO₂ are all processed at similar temperature (950 or 1000 C), the hBN control sample with holes is processed at 700 C. It is very well known by now that hBN is full of different optically active atomic impurities, from UV to NIR (look at these for a quick overview: Aharonovich, I., Englund, D. & Toth, M. Solid-state single-photon emitters. *Nature Photon* 10, 631–641 (2016), Atatüre, M., Englund, D., Vamivakas, N. et al. Material platforms for spin-based photonic quantum technologies. *Nat Rev Mater* 3, 38–51 (2018).), and annealing plays a role in their activation, so what may be happening is that impurities in hBN have a high enough diffusion coefficient at 1000 C to segregate in large quantities at the edges of the crystal, thus also at the graphene/hBN interface, while not having such a mobility at 700 C to be seen by PL.

The authors should thus consider if this hypothesis is realistic, what would be the temperature dependence of the diffusion of impurities in hBN between 700 and 1000 C, look for signs of luminescence at the external edges of the hBN/graphene samples, and if those are dark, ideally processing an hBN control sample with holes in the exact same conditions of the graphene and graphene/hBN samples, and checking for signs of luminescence there.

Reply: During the conversion reaction at high temperature (1000 °C), some defects as an origin of PL may be generated and moved to graphene and h-BN domains or to the interfaces of h-BN and graphene. To check this, we performed PL analysis for each graphene and h-BN monolayer on the SiO₂/Si substrate after annealing in Ar gas flow (50 sccm) at 1000 °C for 5 mins, which is similar to the fabrication of h-BN/graphene in-plane heterostructures. Several points including edges of each monolayer were analyzed for each sample, but PL emission near 400 nm wavelength was not observed (Figure R8). Also, PL was not observed for h-BN monolayer with holes.

Figure R8. PL spectra of graphene (red) and h-BN (blue) monolayers after annealing in Ar gas flow (50 sccm) at 1000 °C for 5 mins, which is similar to the fabrication of h-BN/graphene in-plane heterostructures.

-Concerning the LED

5) As with the previous part, also the LED portion needs a stronger motivation. I find the sentence: “We believe this is a significant achievement in that we have fabricated a GQD-self-emitting device using a single-layer GQD structure isolated by h-BN matrix.” quite tautological. We can rephrase it as: “We have fabricated this new structure and we believe it to be a significant achievement”. This is a belief, not a motivation.

The field is crammed with a plethora of similar works using many different 2D materials-based hybrid structures to make LEDs, the most recent also adding more advanced features such as cavity integration, mechanical flexibility, semi-transparency, large-area etc. (to point at the general concept, one early example is Withers, F., Del Pozo-Zamudio, O., Mishchenko, A. et al. Light-emitting diodes by band-structure engineering in van der Waals heterostructures. *Nature Mater* 14, 301–306 (2015), which however reports an incorrect EQE by a few orders of magnitude -lamentably never corrected- so I would not use it as a benchmark for the EQE specifically).

Similar to the case of the emitters, the introduction does little to position the LEDs within the wider context of the current state-of-the-art, with the only exception of a reference to colloidal GQD-based LEDs, a quite different system, where GQDs are the optically active component. To go to the point: what does set this work clearly apart from the rest? This is not very obvious to me.

Reply: Thank you for pointing it out. We will delete the sentence, “We believe this is a significant achievement in that we have fabricated a GQD-self-emitting device using a single-layer GQD structure isolated by h-BN matrix.”. We are sorry that the sentence made you confused. We agree that GQDs themselves are not active materials in our LED devices, unlike colloidal GQDs. Accordingly, we will revise the sentences related to it in this revised manuscript. As we mentioned before, we would like to focus on the unprecedented PL and its origin at the disordered h-BN/graphene heterojunctions in their in-plane heterostructures. And, the LED device are just a potential application where the heterojunctions of GQDs/h-BN in-plane heterostructures are active components. So, we moved the LED results to supplementary information to clarify the focus of the paper.

6) The EQE is not stellar compared to commercial and state-of-the-art LEDs, but quite in-line with reports from analogous systems. However, I would like to see a vision from the authors. What is their outlook regarding their platform? What is their aim? Where do they see it going? This is completely missing from the conclusions, which right now are just summarizing their findings. A good vision would help strengthen the motivation.

Reply: We agree that the EQE is not high. As we replied to above Point 5, we focus on the unprecedented PL and its origin at the disordered h-BN/graphene heterojunctions in their in-plane heterostructures and the LED device are just an example of potential applications where the heterojunctions of GQDs/h-BN in-plane heterostructures are active components.

In addition, we will have future plan to improve the EQE value of the LED devices. Increasing the amounts of G/h-BN heterojunctions would be the first way to improve the performance. The stacking process mentioned in the paper (Figure 4) is also one way to increase the amounts. Also, we can try the increased density of GQDs by increasing the density of Pt particles during the fabrication of GQDs/h-BN in-plane heterostructures. Lastly, in terms of LED device structure, it is an important method to select electron and hole transport layers matched with the band alignment of the heterojunctions.

7) Can the authors explain in detail how they calculate the external quantum efficiency and which parameters they use?

Reply: The EQE value was calculated by using the following formula. We add the formula in supplementary information.

$$\eta_{EQE} = \frac{\text{Emitted photons}}{\text{Injected Charge}} = \frac{\text{Energy of emitted light} / \text{Energy of one photon}}{\text{Total current} / \text{Charge of one electron}}$$

$$= \frac{\iint_{\lambda, \Omega} \frac{I(\lambda, \Omega) d\lambda (l^2 d\Omega)}{hc/\lambda}}{i_{source}/e} = \frac{\iint_{\lambda, \theta} \frac{I(\lambda, \theta) d\lambda (2\pi l^2 \sin \theta d\theta)}{hc/\lambda}}{i_{source}/e}$$

$$= \frac{\frac{2\pi l^2}{A_{PD}} \iint_{\lambda, \theta} \frac{\Phi_0(\theta) s(\lambda, \theta) d\lambda \sin \theta d\theta}{hc/\lambda}}{i_{source}/e} = \frac{2\pi l^2}{A_{PD}} \frac{e}{i_{source}} \iint_{\lambda, \theta} \frac{\frac{i_{PD}(\theta) s(\lambda, \theta) d\lambda \sin \theta d\theta}{hc/\lambda}}{\int s(\lambda, \theta) R_{PD}(\lambda) d\lambda}$$

PD : photodiode

λ : Wavelength (WL), Ω : Solid angle

$I(\lambda, \theta)$: Radiant intensity at WL reaching the PD (with distance of l) at angle θ

$\Phi(\lambda, \theta)$: Luminous flux at WL reaching the PD (with distance of l) at angle θ

$\Phi_0(\theta)$: Total luminous flux reaching the PD (with distance of l) at angle θ

$s(\lambda, \theta) : (\int s_0(\lambda, \theta) d\lambda = 1)$ = Measured normalized EL spectrum at angle θ

$i_{PD}(\theta)$: PD current at angle θ

$R_{PD}(\lambda)$: Responsivity of PD

l = Distance between the PD and the device

Reviewer #3

This paper is exploring the PL and EL from regions at the interface between graphene and hBN. The authors claim of PL localization in the blue spectral region by the graphene/hBN interface. The ability to obtain PL from 2D materials is an important area, but this paper lacks substantial structural information to draw conclusion. At the moment the authors cannot be sure what is the source of the PL, and can only speculate.

1. The authors have to perform atomic resolution TEM imaging of the grain boundary region between the graphene and hBN to determine the actual structure.

Reply: Please look at our reply to Point 1 of Reviewer #1 where we showed atomic resolved TEM images at boundaries of h-BN and graphene.

2. There is no actual evidence that graphene quantum dots have been produced. The evidence is missing. The authors have to perform some annular dark field scanning transmission electron microscopy to show the carbon regions within the hBN host. Or EELS mapping to show C segregated in the hBN. It could be that there is diffusion of C, B, N in these regions

Reply: The GQD/h-BN structure was already characterized in our previous paper (*Nat. Commun.* **2019**, *10*, 230) where we showed Raman, SEM, XPS, TEM, and EELS data to confirm the formation of GQDs array in the h-BN matrix. In particular, the EELS mapping and spectra were provided for direct evidence of GQDs array in the h-BN matrix (the same data are shown again in Figure R8 below). Based on the EELS mapping and spectra, we confirmed that hBN was converted to GQDs on Pt NPs and B and N are not doped in GQDs in the EELS detection limit.

Figure R8. EELS spectrum of GQD/hBN. **a** Schematics of EELS mapping of GQD/h-BN on Pt NPs/SiO₂. **b**, TEM image of GQD/h-BN on Pt NPs. The white dots are 7 nm Pt NPs. **c-e**, Corresponding EELS mapping images of (c) carbon, (d) boron, and (e) nitrogen, respectively. **f**, Magnified image marked in **d**. The boron signal was not detected at P4 to P7 where the Pt NP exist. Note that the points P5 and P6 with strong signal are due to strong background of Pt signal because we could not completely subtract the strong Pt background. Note that boron signal is absent in P5 and P6 (see panel **g**). In the nitrogen mapping image, the N signal is too low to be detected (see *Nano Lett.* 2013, 13, 1834). **g**, The EELS spectra were obtained at different positions (yellow line, P1 to P10) with 2 nm spatial resolution in **f** by subtracting the background of the Pt signal from the original EELS spectra. The peak for Boron is not detected in P4, P5, P6, and P7, indicating conversion of BN to graphene. Note that GQDs in **c** and **g** are not distinguishable from carbon signal of many adsorbates. This figure is from our previous paper (*Nat. Commun.* 2019, 10, 230).

Overall, without atomic structure details, the paper cannot be published in a high impact journal because the authors cannot know the cause of the PL signals. Without knowing the cause (i.e structure-property correlation), the paper lacks sufficient depth and novelty for publication.

Reply: We showed atomic resolved TEM images and EELS data. Thus, we believe that above data support that the origin of PL is the disordered boundary of h-BN and graphene consisting of 5- and 7-membered rings where the two localized energy states corresponding to the PL energy are found from theoretical results.

REVIEWER COMMENTS

Reviewer #1 (Remarks to the Author):

I appreciate the author's efforts to answer the questions. Revised manuscript with the added figures of the TEM images of grain boundaries and the added physical interpretations looks much more solid. I think the manuscript may be published in Nature communications.

Reviewer #2 (Remarks to the Author):

I believe the authors have improved significantly their manuscript, answered well enough to my questions, and the paper is relevant and well made enough for the audience of Nat Comms to be worth the publication.

For fairness, I only ask that they add the 2 following references to 23-25 (all papers appeared back to back):

-Tonndorf, P. et al. Single-photon emission from localized excitons in an atomically thin semiconductor. *Optica* 2, 347 (2015)

-Srivastava, A. et al. Optically active quantum dots in monolayer WSe₂. *Nature Nanotechnology* 10, 491–496 (2015)

Best Wishes

Reviewer #3 (Remarks to the Author):

The paper has been revised and more data added. The extra data has surely improved the paper, but from my view the same challenge still exists about correlating the structure to properties. The added TEM data shows that the grain boundary region is far from uniform and contains impurity heavy atoms, large nanoholes, various defects and dopants. So many variable structures are present that it is not possible to associate the PL with one particular item in the region. Furthermore the images are not clear enough to actually deduce the exact atomic stitching at the grain boundary region and the authors do not match the experiment to theory, but rather just point and look.

The fact that the authors have published the methods on making graphene quantum dots in hBN before, makes the only novelty the observation of the PL at the interface region. While it is interesting and new, it is not clear that this work has sufficient depth of understanding to warrant high impact publication, it is an observation, rather than a scientific conclusion.

Point by Point Replies to Reviewer's Comments

Reviewer #1

I appreciate the author's efforts to answer the questions. Revised manuscript with the added figures of the TEM images of grain boundaries and the added physical interpretations looks much more solid. I think the manuscript may be published in Nature communications.

Reply: We are grateful to Reviewer #1 for the positive evaluation of our work.

Reviewer #2

I believe the authors have improved significantly their manuscript, answered well enough to my questions, and the paper is relevant and well made enough for the audience of Nat Comms to be worth the publication.

For fairness, I only ask that they add the 2 following references to 23-25 (all papers appeared back to back):

-Tonndorf, P. et al. Single-photon emission from localized excitons in an atomically thin semiconductor. *Optica* 2, 347 (2015)

-Srivastava, A. et al. Optically active quantum dots in monolayer WSe₂. *Nature Nanotechnology* 10, 491–496 (2015)

Reply: We are grateful to Reviewer #2 for the positive comment and suggestion. Accordingly, we add both papers to the reference part of our revised manuscript.

Reviewer #3

The paper has been revised and more data added. The extra data has surely improved the paper, but from my view the same challenge still exists about correlating the structure to properties. The added TEM data shows that the grain boundary region is far from uniform and contains impurity heavy atoms, large nanoholes, various defects and dopants. So many variable structures are present that it is not possible to associate the PL with one particular item in the region. Furthermore the images are not clear enough to actually deduce the exact atomic stitching at the grain boundary region and the authors do not match the experiment to theory, but rather just point and look.

Reply: We basically agree with Reviewer #3 both on the sample state and the TEM image quality. We admit that our image quality might not be the best, but we would still claim that the present TEM images and EELS maps provide minimum necessary information to validate our structure model of the disordered h-BN/graphene heterojunction. Furthermore, we identified from EELS data that the bright spots in our TEM images are Si and Ca atoms possibly coming from the sample transfer process as shown in Fig. R1 below, supporting that our PL does not come from the bright spots.

Also, to exclude possibilities that variable structures such as heavy atoms, large nanoholes, defects, and dopants that might occur during our sample preparation are origins for blue emission in our observation, we investigated energy gaps for variable structures of graphene and h-BN in literature: doped Pt atoms (~ 0.5 eV)¹⁻³, H-terminated edges by large nanoholes (< 1.5 eV for graphene and > 4 eV for h-BN)^{4, 5}, and point defects (1.98 eV for anti-site nitrogen vacancy^{6, 7}, 2.58 eV for nitrogen vacancy^{6, 8}, ~ 4 eV for carbon doping^{8, 9}). There is no paper consistent with our results showing the emission with an energy of 3 eV. The variable structures would show random and broad PL signals that we did not observe in our study. Note that our PL peak shows relatively narrow bandwidth.

Thus, we added above some variable structures in the revised manuscript (8 page).

“Our finding for the unprecedented PL phenomenon at disordered G/BN heterojunction and its origin are different from the previously reported light emission in 2D materials caused by (i) doped Pt atoms²⁰⁻²², (ii) hydrogenated edges^{23, 24} in large nanoholes, (iii) atomic defects like $N_B V_N$ (nitrogen substitution at boron site + nitrogen vacancy)^{25, 26} and carbon substitution at nitrogen sites²⁷ in h-BN, and (iv) point defects in WSe_2 ²⁸⁻³².”

Figure R1. STEM-EELS analysis of h-BN/graphene heterostructure. **a,b.** Annular dark-field (ADF) STEM image showing a boundary between graphene and h-BN domains. **c-e.** EELS

data acquired at the positions labeled as #1, #2 and #3 in (a) and (b). The bright impurity atoms found in our STEM images are identified by EELS as silicon and calcium, probably coming from sample transfer process. Note that (a) and (b) correspond to Fig. 3(d) and Supplementary Fig. 6(a), respectively.

We added the following sentence in the caption of Fig. 3.

“The bright impurity atom found in the red box of (e) is identified as silicon by EELS, probably coming from the sample transfer process.”

We also added the following sentence in the caption of Supplementary Fig. 6.

“The bright impurity atoms found in the red box of (b) are identified as calcium by EELS, probably coming from the sample transfer process.”

The fact that the authors have published the methods on making graphene quantum dots in hBN before, makes the only novelty the observation of the PL at the interface region. While it is interesting and new, it is not clear that this work has sufficient depth of understanding to warrant high impact publication, it is an observation, rather than a scientific conclusion.

Reply: We believe that our work has novelty in that the blue emission arises at the disordered 1D h-BN/graphene heterojunction, which has not been reported so far. This is new physical phenomenon because ordered interfaces or edges do not show the blue emission. It is not limited to graphene quantum dots in h-BN matrix, as shown in Fig. 1 where the blue emission was observed only at the interface of in-plane heterostructures with a large graphene circle of ~20 μm diameter in h-BN matrix. Any disordered h-BN/graphene interface shows such PL. Our calculation results support that the origin of PL is the disordered h-BN/graphene heterojunction, and furthermore show a certain universality in the resonance peak emitted at grain boundaries with arbitrary morphology.

References

1. Üzengi Aktürk, O.; Tomak, M., AuPt_n clusters adsorbed on graphene studied by first-principles calculations. *Physical Review B* **2009**, *80* (8), 085417.
2. Fampiou, I.; Ramasubramaniam, A., Binding of Pt Nanoclusters to Point Defects in Graphene: Adsorption, Morphology, and Electronic Structure. *The Journal of Physical Chemistry C* **2012**, *116* (11), 6543-6555.
3. Xu, D., et al., Theoretical Study of the Deposition of Pt Clusters on Defective Hexagonal Boron Nitride (h-BN) Sheets: Morphologies, Electronic Structures, and Interactions with O. *The Journal of Physical Chemistry C* **2014**, *118* (17), 8868-8876.
4. Lu, Y. H., et al., Effects of edge passivation by hydrogen on electronic structure of armchair graphene nanoribbon and band gap engineering. *Appl. Phys. Lett.* **2009**, *94* (12), 122111.
5. Maruyama, M.; Okada, S., Energetics and Electronic Structure of Triangular Hexagonal Boron Nitride Nanoflakes. *Scientific Reports* **2018**, *8* (1), 16657.
6. Tran, T. T., et al., Quantum emission from hexagonal boron nitride monolayers. *Nature Nanotechnology* **2016**, *11* (1), 37-41.
7. Grosso, G., et al., Tunable and high-purity room temperature single-photon emission from atomic defects in hexagonal boron nitride. *Nature Communications* **2017**, *8* (1), 705.
8. Azevedo, S.; Kaschny, J. R.; de Castilho, C. M. C.; de Brito Mota, F., Electronic structure of defects in a boron nitride monolayer. *The European Physical Journal B* **2009**, *67* (4), 507-512.
9. Bourrellier, R., et al., Bright UV Single Photon Emission at Point Defects in h-BN. *Nano Letters* **2016**, *16* (7), 4317-4321.